# Phylogenetic Analysis of Some Species of the *Anopheles hyrcanus* Group (Diptera: Culicidae) in China Based on Complete Mitochondrial Genomes

**DOI:** 10.3390/genes14071453

**Published:** 2023-07-16

**Authors:** Haowei Dong, Hao Yuan, Xusong Yang, Wenqi Shan, Qiuming Zhou, Feng Tao, Chunyan Zhao, Jie Bai, Xiangyu Li, Yajun Ma, Heng Peng

**Affiliations:** 1Department of Pathogen Biology, College of Basic Medical, Naval Medical University, Shanghai 200433, China; wsnd19951031@163.com (H.D.); xsyang12160809@163.com (X.Y.); springxizhao@163.com (C.Z.); 13761503780@163.com (J.B.); jack_lee@smmu.edu.cn (X.L.); 2College of Naval Medicine, Naval Medical University, Shanghai 200433, China; yhao07@126.com (H.Y.); wenqi_shan@163.com (W.S.); tf850706@163.com (F.T.)

**Keywords:** *Anopheles*, Hyrcanus group, mitochondrial genome, phylogeny

## Abstract

Some species of the Hyrcanus group are vectors of malaria in China. However, the member species are difficult to identify accurately by morphology. The development of sequencing technologies offers the possibility of further studies based on the complete mitochondrial genome. In this study, samples of mosquitoes of the Hyrcanus group were collected in China between 1997 and 2015. The mitochondrial genomes of ten species of the Hyrcanus group were analyzed, including the structure and base composition, codon usage, secondary structure of tRNA, and base difference sites in protein coding regions. Phylogenetic analyses using maximum-likelihood and Bayesian inference were performed based on mitochondrial genes and complete mitochondrial genomes The mitochondrial genome of 10 Hyrcanus group members ranged from 15,403 bp to 15,475 bp, with an average 78.23% (A + T) content, comprising of 13 PCGs (protein coding genes), 22 tRNAs, and 2 rRNAs. Site differences between some closely related species in the PCGs were small. There were only 36 variable sites between *Anopheles sinensis* and *Anopheles belenrae* for a variation ratio of 0.32% in all PCGs. The pairwise interspecies distance based on 13 PCGs was low, with an average of 0.04. A phylogenetic tree constructed with the 13 PCGs was consistent with the known evolutionary relationships. Some phylogenetic trees constructed by single coding regions (such as COI or ND4) or combined coding regions (COI + ND2 + ND4 + ND5 or ND2 + ND4) were consistent with the phylogenetic tree constructed using the 13 PCGs. The phylogenetic trees constructed using some coding genes (COII, ND5, tRNAs, 12S rRNA, and 16S rRNA) differed from the phylogenetic tree constructed using PCGs. The difference in mitochondrial genome sequences between *An. sinensis* and *An. belenrae* was very small, corresponding to intraspecies difference, suggesting that the species was in the process of differentiation. The combination of all 13 PCG sequences was demonstrated to be optimal for phylogenetic analysis in closely related species.

## 1. Introduction

Malaria is an infectious disease transmitted by *Anopheles* mosquitoes that seriously threatens human health. According to the World Malaria Report by the World Health Organization, 241 million cases and 627,000 deaths were reported globally in 2020; both numbers increased compared with 2019. In China, malaria once presented a serious risk to people’s health, causing a large number of illnesses and deaths. After years of effective prevention and control, China was certified malaria-free by the World Health Organization in 2021. However, there is still a large number of imported malaria cases in China every year, with 2673, 1085, and 798 cases in 2019–2021 [1,2,3]. In addition, the *Anopheles* vector mosquitoes are still widespread in China, leading to a high risk of local secondary transmission caused by imported cases. Continued monitoring of the malaria vector *Anopheles* is important to prevent re-emergence of malaria in China [4].

*An. sinensis*, *An. lesteri*, *An. dirus*, and *An. minimus* are the most important malaria vectors in China [5]. The distributions of *An. dirus* and *An. minimus* are relatively limited, while *An. sinensis* and *An. lesteri* have much wider distribution ranges [6]. Both *An. sinensis* and *An. lesteri* belong to the Hyrcanus group. This group comprises 25 member species with valid scientific names, some of which are important malaria vectors [7]. Owing to the similarity in morphology, it is difficult to accurately identify some species in the group. These are cryptic species, similar to other malaria vectors as in the *An. gambiae* complex, the *An. minimus* group, and the *An. funestus* group [8,9,10]. In the Hyrcanus group there are cryptic species such as *An. sinensis*, *An. kleini*, and *An. belenrae* that are unable to be distinguished by morphology and can only be identified by gene sequence [11].

Moreover, the phylogenetic relationships between some member species remain controversial. Mitochondrial gene fragment sequences and internally transcribed spacer (ITS) sequences have been used for phylogenetic analysis of the Hyrcanus group [12,13]. In 2005, Rueda summarized the research results concerning the Hyrcanus group in Korea based on the ITS2 sequences and described two new species discovered by Li et al. named *An. kleini* and *An. belenrae* [14]. Before this, all were considered to be *An. sinensis*. Another example is *Anopheles kunmingensis* and *Anopheles liangshanensis*, which are similar in morphology, and laboratory mating experiments showed no reproductive isolation [15]. However, 13 fixed differences in the ITS2 region were found between the two, supporting the hypothesis that they are independent species [16]. In addition, *An. hyrcanus* and *An. pseudopictus* were considered to be two species, but the results of Ponçon et al. indicated that the genetic distances of the ITS2, COI, and COII sequences between the species are in the range of intraspecific differences [17]. Subsequently, Djadid et al. compared the ITS2 sequence differences between *An. hyrcanus* and *An. pseudopictus* in Iran, and the results also supported the two being synonymous species [18]. The taxonomic status of *An. lesteri* and *An. paralia* is also controversial. Hybrid cross experiments showed that they were genetically compatible, producing viable progeny, and the interspecific genetic distances of the ITS2, COI, and COII sequences were very small, indicating that they were the same species [19].

At present, most of the mitochondrial genome sequences of Hyrcanus group have been reported, including COI, COII, ND5, CytB, and rRNA sequences. Although some phylogenetic issues have been clarified using mitochondrial genome fragment sequences, research on the complete mitochondrial genome of the Hyrcanus group has special value for further exploration [20]. The complete mitochondrial genomes should contribute more to the phylogenetic analysis than gene fragments [21,22]. However, to our knowledge, the complete mitochondrial genome sequences have only reported in a few species [23,24,25].

In this study, we sequenced the complete mitochondrial genomes of ten species in the Hyrcanus group distributed in China using next-generation sequencing (NGS) to characterize the mitochondrial genome of the Hyrcanus group, and we evaluated the significance of the complete mitochondrial genomes in phylogenetic analysis of the closely related *Anopheles* species.

## 2. Materials and Methods

### 2.1. Specimen Collection and Species Identification

A total of ten species in the Hyrcanus group distributed in China were collected; these were *An. belenrae, An. crawfordi, An. junlianensis, An. kleini, An. kunmingensis, An. lesteri, An. liangshanensis, An. peditaeniatus, An. sinensis,* and *An. yatsushiroensis*. The specimens were collected from ten sites in seven provinces in China between 1997 and 2015. Adult mosquitoes were collected by mosquito light traps (Houji Dianzi, Shen Zhen, China) or mosquito aspirators (Table 1). The collected adult mosquitoes were killed with chloroform vapor, dried, and stored separately in 1.5 mL centrifuge tubes with a desiccant. Mosquitoes were brought back to the laboratory and identified based on morphological characteristics as Hyrcanus group species and stored at −80 °C until genome extraction. After extraction of genomic DNA, the species identifications of all samples were confirmed by amplifying the ITS2 sequence using PCR [26]. The reaction mixture is 20 μL containing genomic DNA as templates 10 μL premix Taq (Aidlab, Beijing, China), 0.5 μL forward and reverse primers each, 1 μL genomic DNA and 8 μL ddH_2_O. The PCR reaction was conducted in a ProFlex PCR System (Applied Biosystems, Thermo Fisher Scientific, Waltham, MA, USA), and the cycling conditions were 94 °C for 2 min, followed by 30 cycles of 94 °C/30 s, 45 °C/30 s, 72 °C/30 s, with a final extension at 72 °C for 5 min.

### 2.2. Genomic DNA Extraction, Library Construction, and Next-Generation Sequencing

A single mosquito was placed in a 1.5 mL centrifuge tube and was thoroughly ground after adding liquid nitrogen. Then, the adult mosquito genomic DNA was extracted using a DNeasy Blood Tissue Kit (Qiagen, Hilden, Germany) according to the manufacturer’s instructions. The genomic DNA concentration of the extracted samples was measured by the Qubit method, and OD_260_/OD_280_ was detected by Nanodrop (Thermo Scientific, USA). Genomic DNA samples were stored at −80 °C for further procedures. The DNA was broken into 300–500 bp fragments by the instrument Covaris M220 (Covaris, Woburn, MA, USA). Libraries were then constructed using a TruSeq RNA sample Prep Kit (Illumina, San Diego, CA, USA) and quantified using TBS380 Picogreen (Invitrogen, Carlsbad, CA, USA) reagent. Subsequently, the library was recovered using Certified Low Range Ultra Agarose (Bio-Rad, Richmond, VA, USA). The products were amplified and enriched using a cBot TruSeq PE Cluster Kit v3-cBot-HS (Illumina, San Diego, USA) reagent through PCR reaction, and the cycling conditions were 95 °C for 3 min, followed by 8 cycles of 98 °C/20 s, 60 °C/15 s, 72 °C/30 s, with a final extension at 72 °C for 5 min. Finally, the products obtained were sequenced using a TruSeq SBS Kit (Illumina, San Diego, USA) and in a run of 300 cycles in Illumina Novaseq platform (Illumina, San Diego, USA).

### 2.3. Genomic Assembly and Annotation

The original image data obtained by Illumina sequencing were converted to FASTQ format sequence data through base calling to obtain the original sequencing data file. In total, the raw data reads ranging from 57,068,184 (*An. lesteri*) to 147,214,292 (*An. belenrae*). To ensure the accuracy of subsequent bioinformatics analysis, the Fastp v.19.4 software (https://github.com/OpenGene/fastp/, accessed on 16 May 2022) with default parameters was used to filter the original sequencing data to obtain high-quality data [27]. The specific process is as follows: The adapter sequence was removed from reads. Then, the bases containing non-AGCT at the 5′ end were removed. The ends of reads with lower sequencing quality (sequencing quality value less than Q20) were trimmed. Reads with 10% N content were also removed. Small segments less than 50bp in length after adapter and trimming were discarded. After data filtering, statistics of the sequence reads were counted by cutadapt v1.16 software (http://cutadapt.readthedocs.io/, accessed on 16 May 2022). Using FastQC v.0.11.4 (https://www.bioinformatics.babraham.ac.uk/projects/fastqc/, accessed on 20 May 2022) with the default parameters, the raw sequencing data of each sample were evaluated for quality, including base content distribution and base quality distribution statistics and to further obtain quality control and validation data. The mitochondrial genome reads were extracted with *An. sinensis* mitochondrial genome sequence as reference. The de novo method was used to splice the sequenced read sequences for multiple iterations using the NOVO Plasty (https://github.com/ndierckx/NOVOPlasty, accessed on 20 May 2022) online software [28]. The sequencing depth ranged from 83× (*An. junlianensis*) to 7656× (*An. sinensis*) (Table 1). Based on the de novo assembled mitochondrial genome sequences, gene annotation was performed by the annotate module of mitoZ (https://github.com/linzhi2013/MitoZ) and visualized by the visualize module [29]. Finally, all mitochondrial genome sequences were manually verified by Geneious v.11.0 software.

### 2.4. Overall Analysis of Mitochondrial Genome Sequences

The nucleotide composition of each mitochondrial genome was evaluated by MEGA 11 software [30]. According to the nucleotide composition of each sample genome, the A–T skew and G–C skew of each gene were calculated using the following formulas: AT-skew = (A% − T%)/(A% + T%), and GC-skew = (G% − C%)/(G% + C%). The secondary structures of tRNAs were predicted by the online tool MITOS webserver (http://mitos.bioinf.uni-leipzig.de/index.py, accessed on 8 September 2022) [31]. The relative uniform codon usage (RSCU) of each mitochondrial genome coding region was calculated using MEGA 11 software.

### 2.5. Analysis of Variable Sites in PCGs

Based on previous research reports, six pairs of closely related species in the Hyrcanus group were selected to analyze the variation among sites; these were *An. sinensis* and *An. belenrae*, *An. sinensis* and *An. kleini*, *An. belenrae* and *An. kleini*, *An. kunmingensis* and *An. liangshanensis*, *An. junlianensis* and *An. yatsushiroensis*, and *An. crawfordi* and *An. peditaeniatus*. Mitochondrial genome sequences of another two *An. lesteri* were also obtained to analyze intraspecific variation. All three samples were collected from Sichuan Province, China. (Appendix A). The 13 coding regions of the mitochondrial genome were aligned in pairs using the MEGA 11 software, and the variable sites in nucleotide bases and amino acids were calculated.

### 2.6. Phylogenetic Analysis

The mitochondrial genome PCGs, tRNA, and rRNA coding regions were extracted and summarized using Geneious v.11.0. The gene analysis combination was aligned using the Clustal w algorithm in MEGA 11 software and merged into separate files. Pairwise *p*-distances of species were calculated based on 13 PCGs using MEGA 11 software. The following two methods were used to construct phylogenetic trees based on mitochondrial genes and complete mitochondrial genomes. Under the Akaike information criterion (AIC), JModel Test v2.1.10 was used to select the optimal nucleotide substitution model [32]. Afterward, a phylogenetic tree was constructed using MEGA 11 software based on the maximum-likelihood method with bootstrapping values defined from 1000 repetitions. In addition, Bayesian inference was performed using MrBayes v.3.2.7 in two independent and simultaneous runs established for 1,000,000 generations, with four chains including one cold chain and three hot chains sampled every 500 generations, with a relative burn-in of 25% [33]. The topological structures obtained by the two methods were viewed and visualized with MEGA 11 software, and the graphs were edited with the online tool ITOL (https://itol.embl.de/itol.cgi, accessed on 17 October 2022) [34].

## 3. Results

### 3.1. Overall Composition and Structure of the Mitochondrial Genome

The full-length mitochondrial genomes of ten species members in the Hyrcanus group were obtained by NGS. Among these ten sequences, those of seven species (*An. belenrae* (GenBank accession no. OP311320), *An. crawfordi* (GenBank accession no. OP311321), *An. junlianensis* (GenBank accession no. OP311322), *An. kleini* (GenBank accession no. OP311323), *An. kunmingensis* (GenBank accession no. OP311324), *An. liangshanensis* (GenBank accession no. OP311326) and *An. yatsushiroensis* (GenBank accession no. OP311329)) are reported here for the first time. These mitochondrial genomic DNAs were tightly arranged double-stranded circular molecules with lengths ranging from 15,403 bp (*A. kunmingensis*) to 15,475 bp (*An. liangshanensis*). All mitochondrial genomes contained 13 protein-coding regions, 22 tRNA coding regions, and two rRNA coding regions arranged on the J chain (forward) and N chain (reverse) (Figure 1 and Appendix A).

The percentages of (A + T) content in the mitochondrial genomes of 10 species in *An. hyrcanus* group ranged from 78.14% (*An. junlianensis*) to 78.41% (*An. crawfordi*), and the average (A + T) content was 78.23%. The average (A + T) contents of PCGs, tRNA, and rRNA coding regions were 76.78%, 78.25%, and 81.43%, respectively. The mitochondrial genomes of the ten species in Hyrcanus group had positive A–T skew values, with a mean value of 0.028, and G–C skew values were all negative, with a mean value of −0.156 (Figure 2a–c, Appendix A), close to values for other mosquitoes [35,36,37].

### 3.2. Structural Analysis of PCGs

#### 3.2.1. Composition and Structure of PCGs

The total length of the 13 protein-coding region genes was 11,224 bp; these were ATP6 (681 bp), ATP8 (162 bp), COI (537 bp), COII (685 bp), COIII (787 bp), CYTB (1137 bp), ND1 (945 bp), ND2 (1026 bp), ND3 (354 bp), ND4 (1342 bp), ND4L (300 bp), ND5 (1743 bp), and ND6 (525 bp). The percentages of (A + T) content of the protein-coding region genes were from 76.52% (*An. liangshanensis*) to 76.92% (*An. kunmingensis*). The average AT content of the ND6 gene was the highest (85.83%) in PCGs, while the lowest AT content was 70.09% for the COI gene. All A–T skew values of the protein-coding regions were negative. Most of the protein-coding regions had negative G–C skew values, except in the following cases: all G–C skew values of ND1, ND5, ND4, ND4L genes were positive; the COI gene G–C skew values of *An. crawfordi*, *An. kleini*, *An. kunmingensis*, *An. lesteri*, *An. peditaeniatus* and *An. sinensis* were positive, while the COI gene G–C skew value of *An. yatsushiroensis* was 0; and the CYTB gene G–C skew values of *An. junlianensis* and *An. yatsushiroensis* were positive. It is worth noting that all of the COI genes G–C skew values of the protein-coding regions were relatively small, and the average absolute value of the G–C skew was only 0.006. These compositional characteristics have been reported for Anophelinae and other insects [38,39].

#### 3.2.2. Codon Usage in the Protein-Coding Regions

Among the 13 protein-coding regions, nine genes (ATP6, ATP8, COI, COII, COIII, CYTB, ND2, ND3, and ND6) were located on the J chain (forward), and the remaining four genes (ND1, ND4, ND4L, and ND5) were located on the N chain (reverse). The initiation and stop codons of the protein-coding regions of the ten species in Hyrcanus group were the same. Specifically, ATG was used as the initiation codon of ATP6, COII, COIII, CYTB, ND4, and ND4L; ATT was used as the initiation codon of ATP8, ND1, ND2, and ND6. In addition, TCG, ATA, and GTG were the initiation codons in COI, ND3, and ND5, respectively. For the stop codons, TAA was used as the stop codon of ATP6, ATP8, ND1, ND2, ND3, ND4L, ND5, ND6, and CYTB. However, the single-nucleotide T was used as the stop codon of COI, COII, COIII, and ND4. The recording of incomplete stop codons (T/TA) is common in insects, where the TAA stop codon may be completed by the addition of 3′ A residues to the mRNA after transcription [40,41].

Except for stop codons, all ten species had 3731 codons in the mitochondrial genome PCGs. The relative synonymous codon usage (RSCU) analysis showed that almost all codons were used, except for CUG (L) and AGG (S) codons, which were not used in any of the coding regions. The frequency of codon usage was different among mosquito species. The CUC (L) codon was only used in *An. kleini*, *An. lesteri*, and *An. liangshanensis* with an average RSCU value of 0.01; CCG (P) codons were only used in *An. sinensis*, *An. belenrae*, *An. crawfordi*, and *An. peditaeniatus* with an average RSCU value of 0.03; and CGC(R) codons were only used in *An. liangshanensis*, *An. kunmingensis* and *An. peditaeniatus* with an average RSCU value of 0.07. The GGC(G) codon was unused only in *An. belenrae* and *An. peditaeniatus*, while the GCG(A) codon was unused only in *An. sinensis*.

The use of the third base in synonymous codons favored adenine and thymine (uracil) over guanine and cytosine. For example, the most frequently used codon for leucine was UUA with an average RSCU value of 5.27, while the average RSCU value of UUG, which also codes for leucine, was only 0.32, and another CUG codon was not used at all. In terms of codon usage frequency, the highest was UUA (L, average RSCU 5.27), followed by CGA (R, 3.28), UCA (S, 2.55), UCU (S, 2.54), and GCU (A, 2.21), while the lowest were ACG (T, 0.03), ACU (I, 0.03), and GUC (V, 0.06) (Figure 3). The RSCU value was greater than 1 in all NNA and NNU types. This fixed distribution pattern was similar to those reported in other mosquito species [42].

### 3.3. tRNA Structure Analysis

The full-length tRNAs in the mitochondrial genomes of all ten species of *Anopheles* mosquitoes ranged from 1474 bp to 1476 bp, and the average length of individual tRNA genes ranged from 64 bp to 72 bp. Among the 22 tRNA gene coding regions in ten mitochondrial genomes, 21 predicted tRNA secondary structures that were typical cloverleaf stem-loop structures with four arms and one loop. Only the predicted secondary structure of the tRNA^ser1^ gene was atypical, lacking a DHU arm and instead forming a DHU loop (Appendix A). This pattern has been reported in other insects [36,37].

### 3.4. Differential Site Analysis

Differential site analysis was performed on all mitochondrial genome PCGs between six pairs of cryptic species in the *An. hyrcanus* group, *An. sinensis* and *An. belenrae*, *An. sinensis* and *An. kleini*, *An. belenrae* and *An. kleini*, *An. liangshanensis* and *An. kunmingensis*, *An. yatsushiroensis* and *An. junlianensis*, and *An. crawfordi* and *An. peditaeniatus*. The difference in sites between *An. sinensis* and *An. belenrae* was the least (36, 0.32%). The ratio of transitions to transversions was 33:3. This was followed by *An. junlianensis* and *An. yatsushiroensis* (81, 0.72%, 73:8), *An. kunmingensis* and *An. liangshanensis* (123, 1.10%, 102:21), *An. sinensis* and *An. kleini* (190, 1.69%, 157:33), *An. belenrae* and *An. kleini* (192, 1.71%, 160:32), and *An. crawfordi* and *An. peditaeniatus* (513, 4.57%, 312:201). The comparison between single PCGs showed that some gene sequences were the same, including the ATP6, ATP8, COII, ND3, and ND4L genes of *An. sinensis* and *An. belenrae*, the ND3 and ND4L genes of *An. junlianensis* and *An. yatsushiroensis*, and the ND4L gene of *An. kunmingensis* and *An. liangshanensis*.

PCGs with relatively large differences included ND4 (33, 2.46%, 31:2) and ND1 (26, 2.75%, 21:5) between *An. sinensis* and *An. kleini*, ND4 (32, 2.38%, 30:2) and ND1 (26, 2.75%, 22:4) between *An. belenrae* and *An. kleini*, ND5 (25, 1.43%, 22:3) and ND1 (15, 1.59%, 15:0) between *An. kunmingensis* and *An. liangshanensis*, COI (20, 1.30%, 20:0) between *An. junlianensis* and *An. yatsushiroensis*, and ND5 (83, 4.76%, 52:31) and ATP6 (41, 6.02%, 21:20) between *An. crawfordi* and *An. peditaeniatus*. Differential sites were mostly on the third base of the codon encoding the amino acid, thus not changing the encoded amino acid, while non-synonymous variation in the encoded amino acids was mostly due to the variation in the first bases of the codons (Table 2; Appendix A).

In order to understand the level of intraspecific variation in the mitochondrial genome of *Anopheles*, the PCGs of three mitochondrial genomes of *An. lesteri* were compared. There were total 16 different sites (0.14%) among three samples of *An. lesteri*, showing less variation compared to interspecific variation. The variation sites were distributed in coding gene with five in COI, four in ND4, three in ND2 and ND5, two in ND1, and one in ATP8, COII, and CYTB. No variation site was detected in ATP6, COIII, ND3, ND4L, and ND6.

### 3.5. Phylogenetic Analysis

#### 3.5.1. Genetic Distance

In addition to the ten species in the Hyrcanus group mentioned above, the mitochondrial genomes of four other species of *Anopheles* were also included in the phylogenetic analysis, *An. lindesayi*, *An. barbirostris*, *An. dirus,* and *An. minimus*. The complete mitochondrial genomes of *An. lindesayi* (GenBank accession no. OP324636) and *An. barbirostris* (GenBank accession no. OP324637) were sequenced in our lab. The *An. dirus* (GenBank accession no. JX219731) and *An. minimus* (GenBank accession no. KT895423) belonging to the subgenus *Cellia* were downloaded from the GenBank database. All of the PCGs of 14 *Anopheles* were aligned using MEGA 11 software. Nucleotide genetic distances between species were calculated based on the aligned data. The results showed that the smallest interspecies distance existed between *An. sinensis* and *An. belenrae* with a *p*-distance of 0.003. Other small distances existed between *An. junlianensis* and *An. yatsushiroensis* (*p* = 0.007), *An. kunmingensis* and *An. liangshanensis* (*p* = 0.011), and *An. sinensis* and *An. kleini* (*p* = 0.017). In addition, the pairwise *p*-distances between the subgenus *Cellia* and subgenus *Anopheles* were all larger than the *p*-distances within the subgenera (Table 3).

#### 3.5.2. Phylogenetic Analysis Based on Sequences of 13 PCGs

The J model test was used to analyze the data to select the most suitable nucleotide substitution model; this was determined to be GTR + I + G by the AIC standard test. Based on the sequences of 13 PCGs of the mitochondrial genome, phylogenetic trees were constructed using the maximum-likelihood method (Figure 4a) and the Bayesian method (Figure 4b). The topological shapes of the two phylogenetic trees were completely consistent. The 14 species of *Anopheles* were divided into two clades. The *An. dirus* and *An. minimus* belonging to subgenus *Cellia* together formed a clade. The other clade comprised the subgenus *Anopheles*, including the 10 species in the *An. hyrcanus* group, and *An. lindesayi* and *An. barbirostris*. In the clade of subgenus *Anopheles*, the ten species in the *An. hyrcanus* group clustered together, then clustered with *An. barbirostris*, and finally clustered with *An. lindesayi*. The clustering of the *An. hyrcanus* group was divided into two subgroups. One subgroup contained *An. crawfordi* and *An. peditaeniatus,* and the other contained the other eight species. Among the latter, *An. sinensis* and *An. belenrae, An. liangshanensis* and *An. kunmingensis,* and *An. yatsushiroensis* and *An. junlianensis* were the most closely related pairs (Figure 4).

#### 3.5.3. Phylogenetic Analysis Based on Single or Combined Coding Regions

The phylogenetic trees based on single protein-coding regions were constructed using the maximum-likelihood method as above. The results showed that the topological structures of the phylogenetic trees constructed based on only COI and ND4 genes were consistent with the phylogenetic tree constructed using the 13 PCGs, but the bootstrap values were relatively low (Appendix A,b). Moreover, phylogenetic trees constructed based on other genes were different in topology, for example, COII and ND5 (Appendix A,d). In addition to the sequence of a single protein-coding region, we also selected some highly discriminative coding region combinations (using COI, ND2, ND4, ND5, and ND6) to construct the phylogenetic tree. The results showed that the topological structure of the phylogenetic trees constructed based on the combination of COI + ND2 + ND4 + ND5 genes and ND2 + ND4 genes were consistent with the topological structure of the phylogenetic tree constructed using the 13 PCGs, but the bootstrap values of some branches were low (Appendix A). The phylogenetic trees were also constructed based on the full-length sequence of the tRNA and rRNA coding regions, and the topology was quite different from the phylogenetic tree constructed using the 13 PCGs (Appendix A). These sequences were clearly not suitable for analyzing the phylogenetic relationships among the members of the *An. hyrcanus* group.

#### 3.5.4. Phylogenetic Analysis Based on the Complete Mitochondrial Genome Sequence

Based on the full-length mitochondrial genome sequence, including all coding and non-coding regions, we also constructed a phylogenetic tree, and the topology was consistent with the phylogenetic tree constructed using the 13 PCGs (Appendix A).

## 4. Discussion

In this study, we collected samples of the *An. hyrcanus* group distributed in different regions of China, and the species were identified by morphological and molecular characteristics. After extracting the genomic DNA of single mosquitoes, the complete mitochondrial genome sequences of ten species in the *An. hyrcanus* group were obtained using next-generation sequencing. The complete mitochondrial genomes of seven species were reported for the first time. The results showed that the composition and structure of mitochondrial genomes of ten species in the *An. hyrcanus* group were the same, and their total length and nucleotide composition characteristics were highly similar. In addition, RSCU analysis of protein coding regions showed similar features. These results were consistent with the reported characteristics of mitochondrial genomes in *Anopheles* mosquitoes, confirming the reliability of the sequences determined in this study.

Owing to the high morphological similarity, some species in the Hyrcanus group are taxonomically controversial [43]. Therefore, molecular methods have become an important basis for accurate identification of similar species [44]. At present, the molecular identification markers of species in the Hyrcanus group include internal transcribed spacer 2 (ITS2) and COI and COII genes of the mitochondrial genome. Ma et al. analyzed the phylogenetic relationships of 12 species in the Hyrcanus group based on ITS2 sequences [45]. Zhu et al. discussed the phylogenetic relationships of six species of the Hyrcanus group based on mitochondrial genes COI + tRNA + COII, ATP6 + COIII, ND1, and 16S rRNA genes [46]. Fang et al. studied the phylogenetic relationships of 18 species of the Hyrcanus group based on COI sequences [47]. Zhang et al. compared ITS2 and COII fragments in constructing phylogenetic trees and suggested that ITS2 sequences could be a better molecular marker to distinguish closely related mosquito species [48]. However, the lengths of ITS2 sequence fragments are different, and thus it is difficult to compare them in the reconstruction of phylogenetic relationships. It has also been suggested that the analysis of phylogenetic relationships may lose some information by using fragment reconstruction. The ITS2 sequence can be used to identify some species in the Hyrcanus group. However, the inconsistent length of sequences in various species, large intraspecific variations especially in *Anopheles*, and limited evolutionary information provided by a single gene sequence make it not effective for phylogenetic relationship analysis. Overall, ITS2 is suitable for molecular identification of similar species, while the 13 PCGs of the mitochondrial genome are effective for phylogenetic analysis.

Pairwise nucleotide differences in the mitochondrial genome between *An. sinensis* and *An. belenrae* were very small. In fact, *An. sinensis* and *An. belenrae* were also very similar in ITS2 sequences. Considering their high similarity in morphological and molecular characteristics and overlapping distribution ranges, we believe that *An. belenrae* and *An. Sinensis* were in the process of differentiation. However, *An. kleini* is different in molecular characteristics from *An. sinensis* and *An. belenrae*, and the distribution range is more inclined to high latitudes, supporting it being an independent species.

The differences between *An. junlianensis* and *An. yatsushiroensis*, and *An. kunmingensis* and *An. liangshanensis* were 0.72% and 1.10%, respectively. Some researchers believe that the two groups of *Anopheles* are synonyms [13]. The results of this study partially support this view, as their level of differentiation was between interspecific and intraspecific. Therefore, the two may still be in the process of species differentiation, and more samples should be collected to further clarify the differentiation level. However, the difference between *An. crawfordi* and *An. peditaeniatus* was relatively high at 4.57%. They are clearly two different species.

From the multiple phylogenetic trees constructed above, the topological structure of the phylogenetic tree constructed based on the 13 protein-coding regions was stable and consistent with the known evolution of the Hyrcanus group. Most phylogenetic trees based on single genes of mitochondrial genomes cannot distinguish well the evolutionary relationships between Hyrcanus group species, and are very different from morphological classification results. Therefore, phylogenetic relationship analysis based on single gene should be treated with caution. In addition, tRNA and rRNA coding regions were shown to be unsuitable for phylogenetic analysis. The full-length sequence contains PCGs, tRNA, and rRNA coding regions as well as non-coding regions, leading to the reduction in the discrimination compared with the 13 PCGs evolutionary tree. Certain gene combinations can be selected to construct a phylogenetic tree that is very close to the 13 PCGs phylogenetic tree. However, considering that much basic work is needed to verify the best fragment combination, we believe that 13 PCGs should be the best choice for phylogenetic analysis of the species in the Hyrcanus group [49,50].

However, we were not able to collect all species in the Hyrcanus group distributed in China. For some species, the quality of the extracted genomic DNA was insufficient for next-generation sequencing, partly because of the long storage time. In recent years, the rapid progress of urbanization in China has also led to dramatic changes in the habitat environment of *Anopheles* mosquitoes, making it difficult to collect some species. Moreover, there are gene polymorphisms in the mitochondrial genome, and the variation among *An. hyrcanus* group species is common. Differential loci analysis of three *An. lesteri* samples showed only 16 different sites in the whole mitochondrial genome, which indicated a low intraspecific difference. However, due to sample limitations, one sample per mosquito species was sequenced in this study, except for *An. lesteri*, and intraspecific differences were not considered. In the future, we will strive to collect more species and expand the sample size to further improve the mitochondrial genome data of the species in the Hyrcanus group.

In conclusion, the complete mitochondrial genome sequences of ten species in the Hyrcanus group distributed in China were obtained, seven of which were reported for the first time. The difference in mitochondrial genome sequences between *An. sinensis* and *An. belenrae* was very small, attributed to intraspecies difference, suggesting that the species was in the process of differentiation. The combination of all 13 PCG sequence was shown to be optimal for phylogenetic analysis of closely related species. The results of this study contribute to the accurate identification of the vector species of *Anopheles* and provide scientific basis for the prevention and control of mosquito-borne diseases.

## Figures and Tables

**Figure 1 genes-14-01453-f001:**
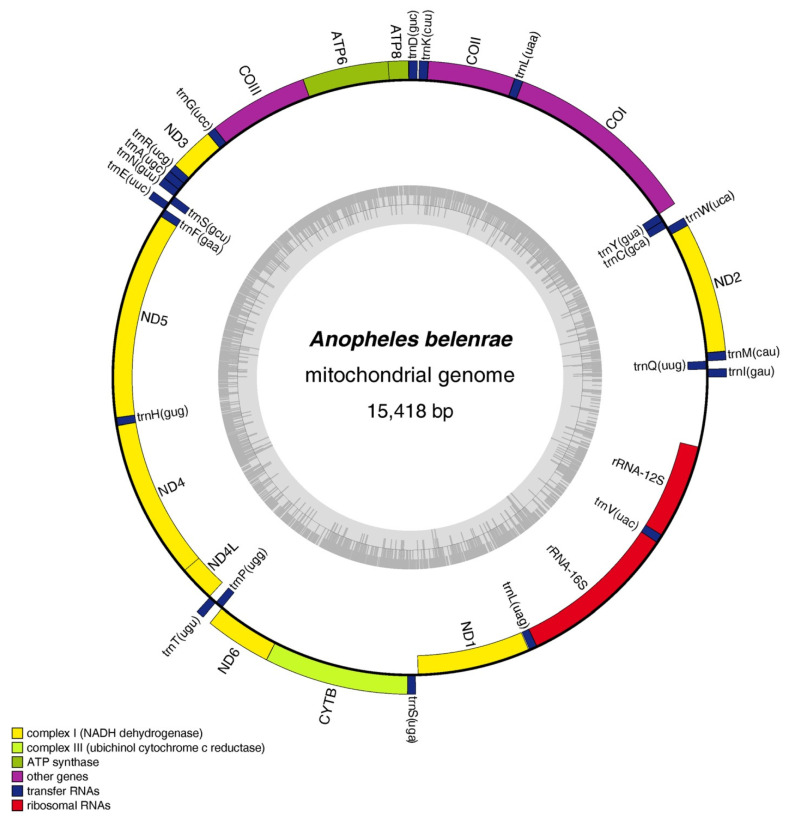
Structures of mitochondrial DNA of *An. belenrae* in the *hyrcanus* group obtained in this study. The mitochondrial genome structures of the other nine *Anopheles* species in this study were similar to that of *An. belenrae* and are listed in Appendix A. The grey inner ring represents the GC content pattern.

**Figure 2 genes-14-01453-f002:**
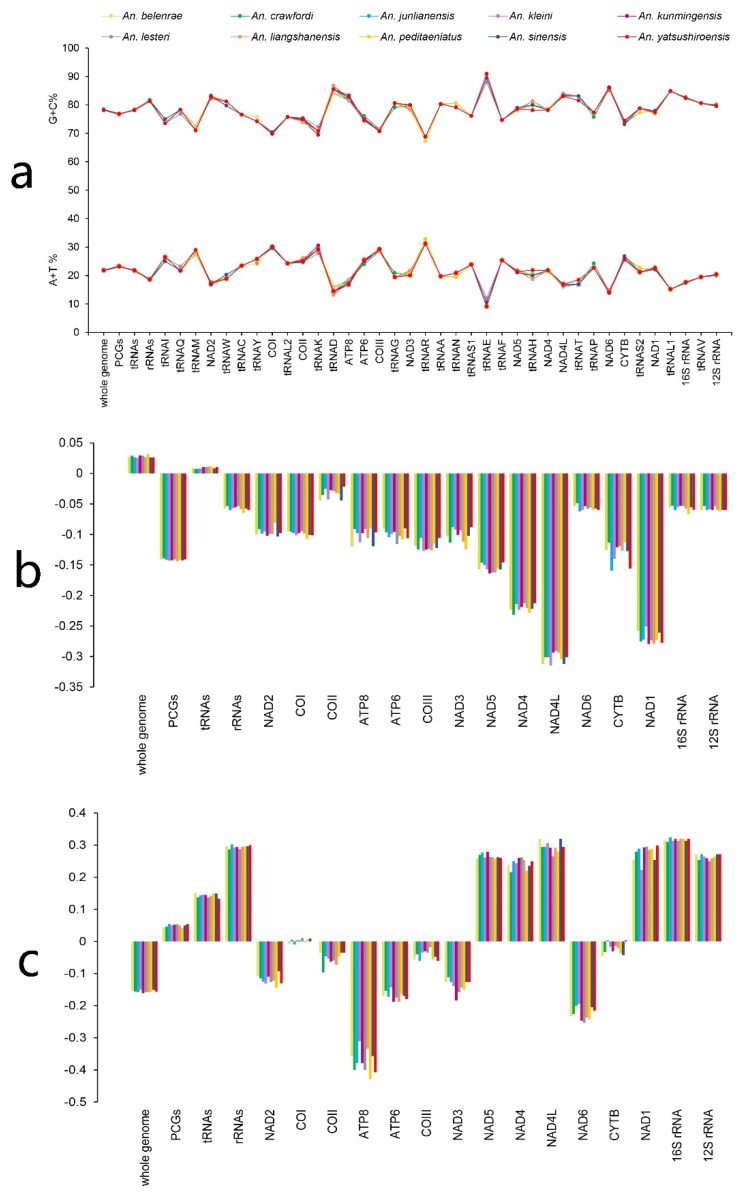
Information on AT/GC contents and AT/GC-skew of the obtained mitochondrial DNAs. (**a**) A + T and G + C contents (%). (**b**) A–T skew. (**c**) G–C skew.

**Figure 3 genes-14-01453-f003:**
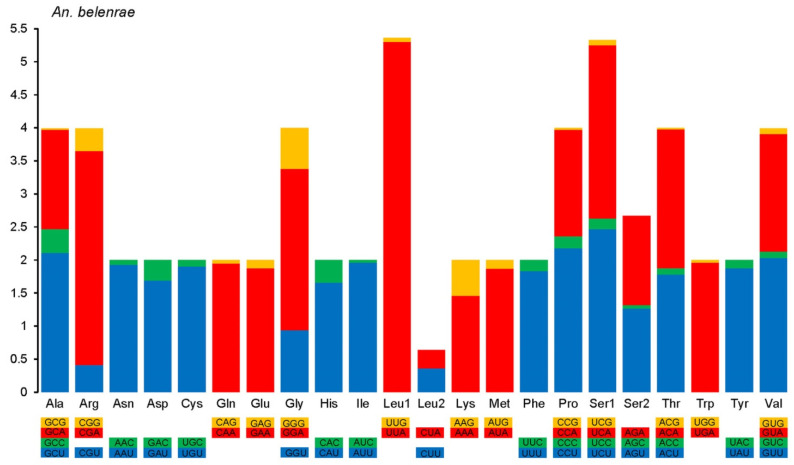
Relative synonymous codon usage (RSCU) of *An. belenrae* mitochondrial DNA. RSCU values are represented on the y-axis, and families of synonymous codons and their respective amino acids are on the x-axis. Colored squares indicate the difference in the third base of the codon: orange for G, red for A, green for C, and blue for U (T).

**Figure 4 genes-14-01453-f004:**
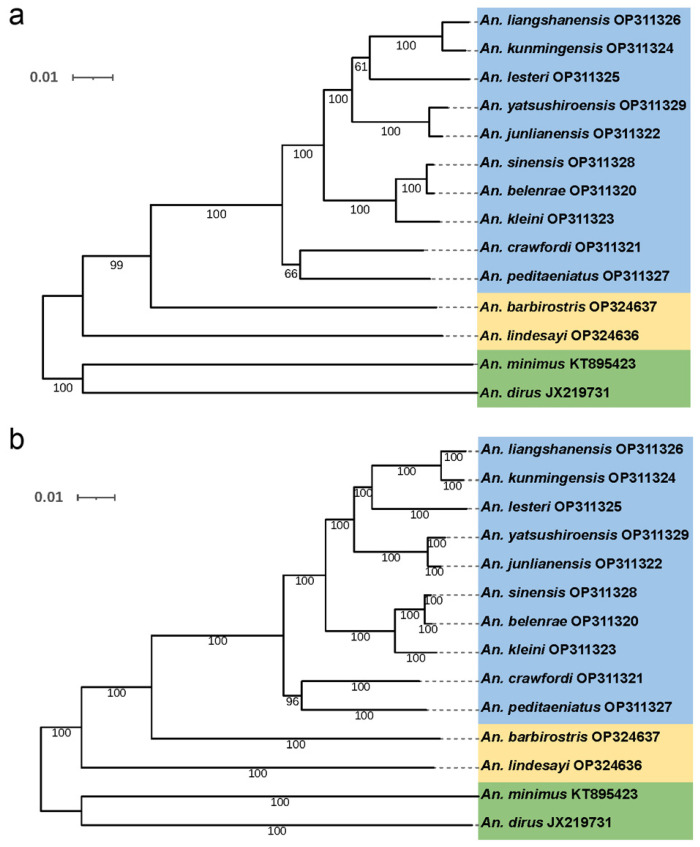
Phylogenetic reconstruction by maximum-likelihood and Bayesian inference based on the 13 PCGs concatenated of the species sequenced in this study and 2 other taxa with data available from GenBank. (**a**) Maximum-likelihood tree. The values of bootstrapping support are shown to the left of the branch point. (**b**) Bayesian inference tree. The values of Bayesian probabilities are shown on the left in each node.

**Table 1 genes-14-01453-t001:** Collection and sequencing information of Hyrcanus group in this study.

Species	Collection Site	Year	Coordinate	Raw Data Reads	Clean Data Reads	Mapped Reads	Mean Depth
*An. belenrae* *	Jining City, Shandong Province, China	1999	116.50° E, 35.32° N	147,214,292	143,104,848	162,947	5755
*An. crawfordi* *	Puer City, Yunnan Province, China	2005	101.08° E, 22.77° N	90,487,882	85,354,442	826,314	7538
*An. junlianensis* *	Junlian City, Sichuan Province, China	1997	104.50° E, 28.15° N	119,378,626	114,520,300	9971	83
*An. kleini* *	Suifenhe City, Heilongjiang Province, China	2018	131.15° E, 44.41° N	87,311,928	87,130,712	163,512	1438
*An. kunmingensis* *	Kunming City, Yunnan Province, China	1997	102.61° E, 24.80° N	88,922,520	86,777,074	475,510	4387
*An. lesteri*	Pujiang City, Sichuan Province, China	2006	103.51° E, 30.20° N	57,068,184	56,937,040	266,795	2553
*An. liangshanensis* *	Zhaojue City, Sichuan Province, China	1997	102.80° E, 28.01° N	80,987,418	80,314,882	714,708	6716
*An. peditaeniatus*	Meilan District, Hainan Province, China	2013	110.51° E, 19.99° N	101,979,708	91,150,574	48,331	396
*An. sinensis*	Jiuzhaigou City, Sichuan Province, China	2015	104.28° E, 33.24° N	121,676,660	120,939,142	1,180,177	7656
*An. yatsushiroensis **	Donggang City, Liaoning Province, China	2000	124.15° E, 39.86° N	117,486,358	114,193,620	162,947	1511

***** means the mitochondrial genome sequence was first reported in this study.

**Table 2 genes-14-01453-t002:** Loci differences between 6 pairs of closely related species in Hyrcanus group.

		ATP6	ATP8	COI	COII	COIII	CYTB	ND1	ND2	ND3	ND4	ND4L	ND5	ND6	PCGs
Total number of sites		681	162	1537	685	787	1137	945	1026	354	1342	300	1743	525	11,224
Single base substitution (transitions: transversions)	SIN and BEL	N	N	7:2	N	2:0	5:0	2:1	7:0	N	5:0	N	3:0	2:0	33:3
	SIN and KLE	10:2	0:1	23:5	4:3	9:0	20:4	21:5	14:2	1:1	31:2	0:1	21:7	3:0	157:33
	KLE and BEL	10:2	0:1	25:5	4:3	9:0	19:4	22:4	17:2	1:1	30:2	0:1	20:7	3:0	160:32
	JUN and YAT	2:1	2:0	20:0	2:2	4:0	8:3	3:0	7:0	N	9:1	N	13:1	3:0	73:8
	KUN and LIA	6:2	1:0	10:3	3:2	7:3	7:1	15:0	12:2	3:0	15:5	N	22:3	1:0	102:21
	PED and CRA	21:20	5:4	35:24	21:11	19:13	40:28	24:20	29:8	6:5	42:28	7:3	52:31	11:5	312:201
Total amino acid number		227	54	512	228	262	379	315	342	118	447	100	581	175	3741
Amino acid mutation (synonymous: non-synonymous)	SIN and BEL	N	N	8:1	N	2:0	5:0	3:0	5:2	N	5:0	N	3:0	2:0	33:3
	SIN and KLE	11:1	0:1	28:0	7:0	8:1	24:0	24:1	14:2	2:0	30:1	1:0	28:0	3:0	180:7
	KLE and BEL	11:1	0:1	29:1	7:0	8:1	23:0	24:1	19:0	2:0	29:1	1:0	27:0	3:0	183:6
	JUN and YAT	3:0	2:0	20:0	4:0	4:0	11:0	3:0	6:1	N	10:0	N	14:0	3:0	80:1
	KUN and LIA	8:0	0:1	13:0	5:0	10:0	8:0	15:0	14:0	3:0	18:2	N	24:1	1:0	119:4
	PED and CRA	39:1	9:0	59:0	32:0	30:2	67:0	42:2	37:0	10:1	70:0	9:1	79:4	14:2	497:13

N means no differences between pairwise data. BEL = *Anopheles belenrae*, CRA = *Anopheles crawfordi*, JUN = *Anopheles junlianensis*, KLE = *Anopheles kleini*, KUN = *Anopheles kunmingensis*, LIA = *Anopheles liangshanensis*, PED = *Anopheles peditaeniatus*, SIN = *Anopheles sinensis*, YAT = *Anopheles yatsushiroensis*.

**Table 3 genes-14-01453-t003:** The pairwise interspecific *p*-distances of mitochondrial DNA of the 14 *Anopheles* species calculated by 13PCGs.

	BEL	CRA	JUN	KLE	KUN	LES	LIA	PED	SIN	YAT	BAR	LIN	DIR
CRA	0.048												
JUN	0.040	0.049											
KLE	0.017	0.049	0.042										
KUN	0.042	0.051	0.035	0.041									
LES	0.041	0.048	0.037	0.042	0.037								
LIA	0.044	0.053	0.036	0.042	0.011	0.036							
PED	0.049	0.046	0.049	0.049	0.050	0.050	0.052						
SIN	0.003	0.048	0.039	0.017	0.041	0.041	0.043	0.048					
YAT	0.040	0.050	0.007	0.042	0.036	0.036	0.038	0.049	0.040				
BAR	0.077	0.075	0.076	0.078	0.078	0.078	0.081	0.075	0.077	0.076			
LIN	0.088	0.086	0.090	0.090	0.090	0.090	0.091	0.089	0.088	0.091	0.090		
DIR	0.096	0.095	0.097	0.099	0.100	0.100	0.101	0.099	0.096	0.097	0.097	0.101	
MIN	0.095	0.093	0.096	0.098	0.097	0.096	0.098	0.096	0.095	0.096	0.090	0.098	0.098

BEL = *Anopheles belenrae*, CRA = *Anopheles crawfordi*, JUN = *Anopheles junlianensis*, KLE = *Anopheles kleini*, KUN = *Anopheles kunmingensis*, LES = *Anopheles lesteri*, LIA = *Anopheles liangshanensis*, PED = *Anopheles peditaeniatus*, SIN = *Anopheles sinensis*, YAT = *Anopheles yatsushiroensis*, BAR = *Anopheles barbirostris*, LIN = *Anopheles lindesayi*, DIR = *Anopheles dirus*, MIN = *Anopheles minimus*.

## Data Availability

All relevant data are within the manuscript and its Supporting Information files. The sequences used and/or analyzed are available in the GenBank (OP311320–OP311329, OP324636–OP324637).

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
