# Peer review of "Phylogenetic Analysis of Some Species of the Anopheles hyrcanus Group (Diptera: Culicidae) in China Based on Complete Mitochondrial Genomes"

_genes, 2023, doi:10.3390/genes14071453_

Round 1

Reviewer 1 Report

The manuscript attempts to bring light to Anopheles hyrcanus systematics using whole mitogenomes. The authors offer a good description of their methodology. A major limitation of the study (acknowledged by the authors) is the small sample size, with a single individual included per species. The work is overall well done and the manuscript has only minor language issues.

Using less conventional taxa, such as “species group” or “species complex” is perhaps confusing for some authors seeking to abide conventions, since the same usage is pervasive in literature.

Please change “Anopheles hyrcanus group” to “Hyrcanus group” throughout the text. In the title change it to “Anopheles hyrcanus sensu lato”.

Materials and Methods:

Table 1 headers: “Raw” instead “Row” , “Mapped reads” instead “Mapping reads”

line 175: change “summarized” and “fragment analysis combination”; phrasing is confusing.

line 179: Replace “gene fragments” with “genes” ; they are the fragments of whole mitogenomes, but complete genes nonetheless.

Results:

Figure 1 and Figure S1: What represents the grey ring placed inside the annotated mitogenome(s)?

Figure 2a: Please indicate precisely on the plot which graph is A+T% and which is G+C%

line 238: Replace “fragments” with “genes” and change throughout the text (see for example line 365: “COI and ND4 fragments”). The most common practice to date is the barcoding of partial genes  (fragment), therefore this wording can cause confusion.

Discussion:

lines 389-390: Seven species’ mitogenomes are being reported for the first time - you could indicate them with an “ * “ or other symbol in Table 1 and caption.

lines 413 - 415: What do the authors mean to say with “ITS2 is more suitable for molecular identification […] while the 13PGGs […] are suitable for phylogenetic analysis? Do they refer to the taxonomic resolution afforded by each, with ITS2 better for fine-scale resolution among sister taxa and 13 PGGs better for deeper phylogenetic nodes? Please clarify

lines 441 and 443: please change “standard evolutionary tree” and clarify what you mean by it. Since a phylogenetic tree is inherently a hypothesis, this does not make sense.

The generation and analysis of whole mitogenomes still is an approach for a minority of studies. Many more studies of mosquito molecular taxonomy have used fragments of genes or single barcodes. 

Could the authors discuss briefly how do the genetic distances within the Hyrcanus group compare to patterns of divergence inside other mosquito species groups? Are reports based on mitogenome PCGs available or these patterns have only been analysed by sequencing of single/partial markers? 

Several expression or words used frequently are confusing in the best case. Suggestions for rectification were provided in the review report.

Author Response

Comment 1

Using less conventional taxa, such as “species group” or “species complex” is perhaps confusing for some authors seeking to abide conventions, since the same usage is pervasive in literature.

Please change “Anopheles hyrcanus group” to “Hyrcanus group” throughout the text. In the title change it to “Anopheles hyrcanus sensu lato”.

 Response: Thank you for your kind suggestions. We have changed the expression of these words in the text.

Comment 2

Materials and Methods:

Table 1 headers: “Raw” instead “Row” , “Mapped reads” instead “Mapping reads”

 line 175: change “summarized” and “fragment analysis combination”; phrasing is confusing.

 line 179: Replace “gene fragments” with “genes” ; they are the fragments of whole mitogenomes, but complete genes nonetheless.

 Response: Thank you for your kind suggestions. We have changed these inappropriate expressions in this paper.

Comment 3

Results:

Figure 1 and Figure S1: What represents the grey ring placed inside the annotated mitogenome(s)?

Response: Thank you for your kind suggestions. The grey ring placed inside the annotated mitogenome(s) represents the GC content graph (inner ring). We have add it to the diagram.

Figure 2a: Please indicate precisely on the plot which graph is A+T% and which is G+C%

Response: Thank you for your kind suggestions. We have marked A+T skew and G+C skew clearly in Figure 2a.

line 238: Replace “fragments” with “genes” and change throughout the text (see for example line 365: “COI and ND4 fragments”). The most common practice to date is the barcoding of partial genes (fragment), therefore this wording can cause confusion.

Response: Thank you for your kind suggestions. We have changed the expression of "fragments" in the paper to "genes".

Comment 4

Discussion:

 lines 389-390: Seven species’ mitogenomes are being reported for the first time - you could indicate them with an “ * “ or other symbol in Table 1 and caption.

Response: Thank you for your kind suggestions. We have indicated these species with an “ * ” in Table 1.

lines 413 - 415: What do the authors mean to say with “ITS2 is more suitable for molecular identification […] while the 13PGGs […] are suitable for phylogenetic analysis? Do they refer to the taxonomic resolution afforded by each, with ITS2 better for fine-scale resolution among sister taxa and 13 PGGs better for deeper c nodes? Please clarify

Response: Thank you for your kind suggestions. The ITS2 sequence can be used to identify some species in the Hyrcanus group. However, the inconsistent length of sequences in various species, large intraspecific variations especially in Anopheles, and limited evolutionary information provided by a single gene sequence make it not suitable for studying phylogenetic relationships. So, maybe ITS2 is a good choice as DNA barcoding, but not suitable for phylogenetic analysis. We revised the sentence as below:

The ITS2 sequence can be used to identify some species in the Hyrcanus group. However, the inconsistent length of sequences in various species, large intraspecific variations especially in Anopheles, and limited evolutionary information provided by a single gene sequence make it not effective for phylogenetic relationship analysis. Overall, ITS2 is suitable for molecular identification of similar species, while the 13 PCGs of the mitochondrial genome are effective for phylogenetic analysis.

lines 441 and 443: please change “standard evolutionary tree” and clarify what you mean by it. Since a phylogenetic tree is inherently a hypothesis, this does not make sense.

Response: Thank you for your kind suggestions. We have used absolute terms here which may cause confusion. Here we mean that the topological structure of phylogenetic maximum likelihood tree based on 13 PCGs corresponds well with traditional morphological classification. We have changed description in the paper.

The generation and analysis of whole mitogenomes still is an approach for a minority of studies. Many more studies of mosquito molecular taxonomy have used fragments of genes or single barcodes. 

Could the authors discuss briefly how do the genetic distances within the Hyrcanus group compare to patterns of divergence inside other mosquito species groups? Are reports based on mitogenome PCGs available or these patterns have only been analysed by sequencing of single/partial markers? 

Response: Thank you for your kind suggestions. Based on previous literature, the genetic distances within the Hyrcanus group vary from 0.007 to 0.07 (K2P-distance), while other mosquito species groups such as Neotropical Albitarsis Group shows a genetic distance between 0.01 to 0.3 based on COI gene fragment.

As you mentioned, there are few studies that obtain and classify the complete mitochondrial genome coding region, and many studies are only descriptive reports. At present, the molecular identification markers of close relatives or recessive species are mainly based on a certain fragment, and COI, COII fragment and ITS2 fragment of mitochondrial genome are commonly used in Anopheles mosquitoes. But recently, a growing number of articles are focusing on other segments of the mitochondrial genome and analyzing their role in rebuilding phylogenetic relationships, such as the ND4 gene.

The purpose of this study is to report the mitochondrial genome sequence and structure of 7 Anopheles species for the first time. Based on the existing data, we attempt to analyze the role of complete mitochondrial genome and complete fragments of each coding region in distinguishing the species that are very close related, which can provide basic data reference for the molecular identification of Anopheles hyrcanus group.

Reference

Abuelmaali SA, Jamaluddin JAF, Allam M, et al. Genetic Polymorphism and Phylogenetics of Aedes aegypti from Sudan Based on ND4 Mitochondrial Gene Variations. Insects 2022; 13: 1144.

Motoki MT, Linton Y-M, Conn JE, Ruiz-Lopez F, Wilkerson RC. Phylogenetic Network of Mitochondrial COI Gene Sequences Distinguishes 10 Taxa Within the Neotropical Albitarsis Group (Diptera: Culicidae), Confirming the Separate Species Status of Anopheles albitarsis H (Diptera: Culicidae) and Revealing a Novel Lineage, Anopheles albitarsis J. J Med Entomol 2021; 58: 599–607.

Ali RSM, Wahid I, Saingamsook J, et al. Molecular identification of mosquitoes of the Anopheles maculatus group of subgenus Cellia (Diptera: Culicidae) in the Indonesian Archipelago. Acta Trop 2019; 199: 105124.

Reviewer 2 Report

The authors presented a very interesting and scientifically sound study on phylogenetic analysis of cryptic Species of the Anopheles hyrcanus group based on mitochondrial genomes. After careful consideration, I recommend the publication of the study after the correction of minor issues pesented below:

M and M

In section 2.1 the species name must be italicized

A figure showing a map of the collection sites would be very useful

Why only An. lesteri was used to check for itraspecific variation?. The authors should explain the rationale for that

Results

In figure 1, I assume that the gray bars in the center is a coverage bar plot. The authors should add this information in the caption of the figure.

In line 329, the authors state that ”smallest interspecies distance existed between An. sinensis and An. belenrae with a p value of 0.003”. The authors should rephrase that because p-distance can be misunderstood as p-value for statistics.

Author Response

Reviewer #2

Comment 1

In section 2.1 the species name must be italicized

Response: Thank you for your kind suggestions. The species name in this section have been italicized.

A figure showing a map of the collection sites would be very useful.

Response: Thank you for your kind suggestions. We have added a schematic map (Fig. S1) of the collection sites for this research.

Comment 2

Why only An. lesteri was used to check for itraspecific variation?. The authors should explain the rationale for that.

Response: Anopheles lesteri is widely distributed in China and has a certain representative name in the Hyrcanus groups species. It has been reported that the genetic distance between different geographical populations of Anopheles lesteri is large, and there may be cryptospecies. In addition, the sampling time span of this study is long, according to limited specimens, we chose Anopheles lesteri to analyze the size of intraspecific differences and explore the possibility of cryptospecies.

Along with the one sample sequenced already, differential site analysis of the coding regions of three An. lesteri was calculated. The result showed 16 base differences in these three samples. These results indicated that the intraspecific variations were relatively small, reflecting the stability of sequencing. In this study, the smallest interspecific variation was between An. belenrae and An. sinensis with 36 bases difference. We hope to analyze and discuss the intraspecific differences and variation in mitochondrial genome sequence of An. hyrcanus group in further studies.

Comment 3

Results

In figure 1, I assume that the gray bars in the center is a coverage bar plot. The authors should add this information in the caption of the figure.

Response: Thank you for your kind suggestions. The grey ring placed inside the annotated mitogenome(s) represents the GC content graph (inner ring). We have add it to the diagram.

In line 329, the authors state that ”smallest interspecies distance existed between An. sinensis and An. belenrae with a p value of 0.003”. The authors should rephrase that because p-distance can be misunderstood as p-value for statistics.

Response: Thank you for your kind suggestions. We have changed the expression of “p value” to “p distance”.

Reviewer 3 Report

The authors exploited cryptic Species of the Anopheles hyrcanus group inhabiting chinese areas to produce a phylogenetic reconstruction by means of sequencing mitochondrial genomes. The work seems well done, methods are adequate but I have missed some explanations on how the work provides a basis for malaria control as stated by the authors. For example,  "This group comprises 25 member species with valid scientific names, some of which are important malaria vectors". If I am not mistaken, do the authors mean that some Anopheles species are not malaria vectors? I think it is important to clarify this point to the readers. In this sense, map showing locations of collections could be added to the manuscript and, ideally, temperature and altitude from the collection points could give a better idea of species habitat. This is particularly important for a manuscript data that, according to the authors, are intended to help in the mosquito control.

Sequencing results showed minor differences among Anopheles mitogenomes that, perhaps I missed the point, were intended to overcome morphological diagnostics as they are cryptic species. The authors may want to discuss in the text that the ITS2, despite length divergence, could be a good candidate to species discrimination by using primer pairs in PCR.

Overall, the text can be understood in its present form.

Author Response

Reviewer #3

Comment 1

The work seems well done, methods are adequate but I have missed some explanations on how the work provides a basis for malaria control as stated by the authors. For example, "This group comprises 25 member species with valid scientific names, some of which are important malaria vectors". If I am not mistaken, do the authors mean that some Anopheles species are not malaria vectors? I think it is important to clarify this point to the readers.

Response: Thank you for your kind suggestions. Anopheles mosquitoes are the vectors that transmit malaria. There are more than 400 species of Anopheles mosquitoes worldwide, 67 species were reported naturally infected by plasmodium, and 41 species were believed playing important roles in malaria transmission. The main vectors of malaria transmission in China include An. sinensis, An. lesteri, An. minimus and An. dirus, in which An. sinensis, An. lesteri belong to the Hyrcanus group. An. kleini, Anopheles pullus, An. belenrae, An. nigerrimus, An. peditaeniatus and An. hyrcanus from the Hyrcanus group were considered potential vectors in malaria transmission.

Reference

Sinka M E, Bangs M J, Manguin S, et al. A global map of dominant malaria vectors[J]. Parasite Vector, 2012,5(1): 69.

Comment 2

In this sense, map showing locations of collections could be added to the manuscript and, ideally, temperature and altitude from the collection points could give a better idea of species habitat. This is particularly important for a manuscript data that, according to the authors, are intended to help in the mosquito control.

Response: Thank you for your kind suggestions. A schematic map (Fig. S1) of the collection sites for this research was added.

Comment 3

Sequencing results showed minor differences among Anopheles mitogenomes that, perhaps I missed the point, were intended to overcome morphological diagnostics as they are cryptic species. The authors may want to discuss in the text that the ITS2, despite length divergence, could be a good candidate to species discrimination by using primer pairs in PCR.

Response: Thank you for your kind suggestions. The ITS2 sequence can be used to identify some species in the Hyrcanus group. However, the inconsistent length of sequences in various species, large intraspecific variations especially in Anopheles, and limited evolutionary information provided by a single gene sequence make it not effective for phylogenetic relationship analysis. Overall, ITS2 is suitable for molecular identification of similar species, while the 13 PCGs of the mitochondrial genome are effective for phylogenetic analysis.
